# Antibacterial Efficacy and Surface Characteristics of Boron Nitride Coated Dental Implant: An In-Vitro Study

**DOI:** 10.3390/jfb14040201

**Published:** 2023-04-06

**Authors:** Anjali Raval, Naveen S. Yadav, Shweta Narwani, Kirti Somkuwar, Varsha Verma, Hussain Almubarak, Saeed M. Alqahtani, Robina Tasleem, Alexander Maniangat Luke, Sam Thomas Kuriadom, Mohmed Isaqali Karobari

**Affiliations:** 1Department of Prosthodontics Crown Bridge and Implantology, Peoples Dental Academy, Peoples University, Bhopal 462037, Madhya Pradesh, India; 2Department of Diagnostic Sciences & Oral Biology, College of Dentistry, King Khalid University, Abha 62529, Saudi Arabia; 3Department of Prosthetic Dentistry, College of Dentistry, King Khalid University, Abha 62529, Saudi Arabia; 4Department of Prosthodontics, College of Dentistry, King Khalid University, Abha 62529, Saudi Arabia; 5College of Dentistry, Centre of Medical and Bio-Allied Health Sciences Research, Ajman University, Al-Jruf, Ajman P.O. Box 346, United Arab Emirates; 6Department of Restorative Dentistry & Endodontics, Faculty of Dentistry, University of Puthisastra, Phnom Penh 12211, Cambodia; 7Department of Conservative Dentistry & Endodontics, Saveetha Dental College & Hospitals, Saveetha Institute of Medical and Technical Sciences University, Chennai 600077, Tamil Nadu, India

**Keywords:** boron nitride, antibacterial properties, streptococcus mutans, fusobacterium nucleatum, periimplantitis, dental implants

## Abstract

This in vitro study evaluated bacterial cell proliferation and biofilm adhesion on titanium discs with and without antibacterial surface treatment to reduce the chances of peri-implant infections. Hexagonal boron nitride with 99.5% purity was converted to hexagonal boron nitride nanosheets via the liquid phase exfoliation process. The spin coating method was used for uniform coating of h-BNNSs over titanium alloy (Ti6Al4V) discs. Two groups of titanium discs were formed: Group I (*n* = 10) BN-coated titanium discs and Group II (*n* = 10) uncoated titanium discs. Two bacterial strains, Streptococcus mutans (initial colonizers) and Fusobacterium nucleatum (secondary colonizers), were used. A zone of inhibition test, microbial colony forming units assay, and crystal violet staining assay were used to evaluate bacterial cell viability. Surface characteristics and antimicrobial efficacy were examined by scanning electron microscopy with energy dispersion X-ray spectroscopy. SPSS (Statistical Package for Social Sciences) version 21.0 was used to analyze the results. The data were analyzed for probability distribution using the Kolmogorov-Smirnov test, and a non-parametric test of significance was applied. An inter-group comparison was done using the Mann-Whitney U test. A statistically significant increase was observed in the bactericidal action of BN-coated discs compared to uncoated discs against *S. mutans*, but no statistically significant difference was found against *F. nucleatum*.

## 1. Introduction

A dental implant’s success or failure can be greatly influenced by both osseointegration, which occurs when an implant integrates with the bone, and bacterial aggregation surrounding the implant [1] Even when antibiotics are frequently and systemically delivered in a clean environment, implant-related infections can have serious consequences [1]. Due to their partial contact with the jawbone and gums, dental implants are constantly exposed to oral microorganisms. Additionally, the pathogenesis of infections involving biomaterials is significantly influenced by the adhesion and colonization of bacteria. Integration of the dental implant material into the surrounding bone and connective tissue is important for the long-term stability of the implant [1,2,3]. Several surface treatments can reduce the incidence of implant-associated infection [1]. 

Titanium alloy (Ti6Al4V) is the most commonly used implant material due to its favorable mechanical properties and biocompatibility, high corrosion resistance, lack of cell toxicity, and the minimal inflammatory response it causes in the tissues around the implant. Positive regulation of biological processes cannot be fully achieved with titanium alloy since it cannot cause bone apposition. Additionally, because biological processes at the bone-implant interface proceed more slowly on smooth surfaces, titanium alloy surfaces are treated to improve their surface characteristics, shorten healing time, and increase bone-implant contact areas and osseointegration [2].

In boron nitride (BN), boron and nitrogen atoms have a honeycomb structure resembling that of graphene. Specifically, strong sp2 covalent in-plane bonding and weak van der Waals forces between layers make up the structure [3,4,5,6,7,8,9]. In a recent theoretical investigation using molecular dynamics simulations, it was shown how BN nanoflakes interacted with model cell membranes: they strongly attracted to the polar head groups in bilayer lipids [3,9,10]. It is believed that BN and its derivatives have a significant potential for use in biomedical applications. Recent investigations showed that BN has the potential to eliminate greasy substances and chemicals (such as organic solvents and dyes) from water. They have also been used in DNA/RNA self-assembly and the administration of anticancer drugs [3,7,8]. Optoelectronic nanodevices, multifunctional composite materials, hydrogen accumulators, and insulating substrates are already made with BN nanoparticles [10].

An earlier in vitro investigation found that BN/LDPE (low-density polyethylene) composite physically interacted with the bacterial cellular membrane, causing permanent physical damage. Specifically, it showed bacteriostatic activity against *Escherichia coli* and *Staphylococcus aureus* [3,4]. BN nanoparticles have also been utilized in the creation of electrospun hybrid nanostructure BN/Ag for antibacterial purposes [5]. At the site of physical cell damage, pristine and antibiotic-loaded nanosheet-based boron nitride films induce oxidative stress and may be a promising platform to suppress bacterial and fungal infections [6]. Boron nitride nanotubes were added to adhesive resin materials to improve their physical and chemical characteristics as well as increase the number of minerals deposited on their surface [11,12]. Boron nitride nanoplatelets can be used as a biocompatible reinforcement to improve the physical properties of dental ceramics and self-cured PMMA (poly (methyl methacrylate)) materials [13,14].

Antibiotics have been used to treat bacterial infections for decades, but they are unable to eliminate the microorganisms found in biofilms, which are a primary source of infection on implant surfaces. When biofilms develop on an the implant’s surface, they create a barrier that can protect bacteria from antibiotics and other antibacterial agents. However, the overuse of antibiotics not only encourages the proliferation of microorganisms resistant to them but also decreases their effectiveness. Therefore, to inhibit bacterial adherence and biofilm formation, an efficient surface modification with an antibacterial action is needed [14,15].

BN and its derivatives, including BN nanotubes, appear to be less toxic and more biocompatible [15,16,17]. BN nanotubes’ interactions with various cells demonstrated very low levels of cytotoxicity [15,18,19]. Similarly, plasma treatment of BN nanotube films improved cell attachment, and BN nanotube films increased the proliferation of human mammary cells [20,21,22,23,24,25,26,27,28,29,30,31,32]. suggesting that these materials may have a significant potential for use in a variety of implant technologies. 

The current study aimed to evaluate the antibacterial effect and surface topography of boron nitride-coated implants to prevent pellicle formation and bacterial colony formation, thereby reducing the chances of peri-implantitis. In this manner, the quality of implants may be enhanced for future clinical implications.

## 2. Materials and Methods

This experimental in vitro study compared the antibacterial properties of BN-coated TI discs to uncoated TI discs for dental applications. A total of 20 samples (10 titanium alloy discs and 10 boron nitride-coated titanium alloy discs) were taken. A schematic representation of the experimental process is shown in Figure 1.

### 2.1. Preparation of Bn Solution

Hexagonal boron nitride was purchased (Vedayukt India Private Limited, Jharkhand, India) with 99.5% purity in powder form. The boron nitride powder was mixed in a solvent containing a mixture of isopropyl alcohol (IPA) and de-ionized water (DI) [Himedia] to produce h-BNNSs (hexagonal boron nitride nanosheets) in large quantities by ultrasonication in a 3:7 ratio (ultrasonic isolating chamber, Lark, India) After that, the liquid phase exfoliation method was used to synthesize h-BNNSs. The exfoliation of h-BN is caused by electron-deficient boron atoms, which result from the Lewis acid-base interaction mechanism. The prepared solution was kept in in 6 test tubes in a centrifuge (REMI PR-24, New York, NY, USA). The first centrifugation was done at 1000 rpm for 10 min, followed by 3000 rpm for 10 min, followed by 5000 rpm for 10 min. Thus, h-BNNSs were achieved.

### 2.2. Coating Procedure

Samples of the Grade 5 Ti6Al4V titanium alloy discs (Bhagyashali Metal Private Limited, Maharashtra, India, with dimensions of 10 mm in diameter and 2 mm in thickness) were taken. Discs were ultrasonically cleaned for 2 min. 

The spin coating method was used to add a uniform coating of h-BNNSs over the titanium discs. First, h-BBNSs solution was dropped over each disc with a micropipette at a speed of about 500 rpm. For each disc, 5 cycles were done at 500 rpm for 1 min each. At a fixed rate, a stage of substrate spinning takes place, and the behavior of the fluid is dominated by viscous forces, while that of the coating is dominated by solvent evaporation. Discs were air-dried for 1.5 h and kept in a petri dish over the slide to prevent dust contamination.

### 2.3. Surface Characterization

Samples were examined under a scanning electron microscope (SEM, ZEISS, Jena, Germany) to both obtain coating thickness and to ascertain the impact of BN coating on roughness and wettability; these measurements were conducted on both the uncoated and BN-coated implants.

The Micro Scratch Tester (CSM instruments, Neuchatel, Switzerland) was used to evaluate the BN films’ quality of adhesion. Starting at 0.5 N, the load was gradually raised to 30 N. A 100 m-radius Rockwell diamond indenter tip was employed. When the indenter moved at a rate of 4 mm per minute, a scratch of two millimeters was created. 59.94 N min1 was the loading rate.

Surface roughness was measured using profilometer equipment, which also provides the difference between a surface’s high and low points. During profilometer operation, a diamond stylus was moved vertically while in contact with a sample. The diamond stylus was then moved laterally across the sample for a predetermined distance at a predetermined contact force. Outcomes were assessed using the Image Plus program.

### 2.4. Bacterial Strains

*Streptococcus mutans* [Strain: MTCC 890] and *Fusobacterium nucleatum* [Strain: ATCC 25586] strains were purchased from the Institute of Microbial Technology (Chandigarh, India) and American Type Culture Collection [ATCC, Manassas, Virginia, USA], respectively, for use in the biofilm formation.**Incubation:**Specimens were divided into these two groups:Group I: BN-coated titanium discs [BN],Group II: Uncoated titanium discs [Control].

BHI (brain heart infusion) broth was prepared to inoculate *Streptococcus mutans* [Strain: MTCC 890] and *Fusobacterium nucleatum* [Strain: ATCC 25586] and then incubated (incubator, Yamto, Japan). The sample groups were divided into coated and uncoated BN for 24–48 h of analysis. The discs were sterilized by dipping them in 70% ethanol (Himedia, Mumbai, Maharastra, India) for 1 min. After sterilization, the samples were placed in the broth medium (Himedia, Mumbai, Maharastra, India) for incubation and analysis.

### 2.5. Zone of Inhibition Test

The antimicrobial activity of both BN-coated and control disc samples against *Streptococcus mutans* and *Fusobacterium nucleatum* were tested using the disc diffusion method. About 100 μL of pre-cultured test organisms were spread onto the agar plates, and then the discs were put on the agar plates. Bacterial plates of *S. mutans* were incubated for 24 h at 37 °C, whereas *Fusobacterium nucleatum* plates were incubated at 37 °C for 24 h with 85% N_2_, 5% CO_2_, and 10% H_2_. The zone of inhibition was measured and tabulated. 

### 2.6. Biofilm Adhesion Assay

One set of samples was used for colony counting after incubation, whereas another set was used for crystal violet staining and a UV spectrophotometer test.

Microbial colony forming units (CFUs) assay: The specimens were placed in BHI agar plates. After being incubated for 48 h, the bacteria were counted (Colony Counter, Yamto, Japan). For BN-coated discs bacterial concentration was taken as 1 × 10^3^; for uncoated control discs, it was taken as 1 × 10^6^.

Crystal Violet staining assay: PBS (phosphate buffer saline, Himedia) was autoclaved and stored. The culture broth was carefully decanted, and the disc was given two minutes to dry. Following that, a few drops of crystal violet (Himedia) was added, and the mixture was left alone for five minutes. Next, the disc samples were rinsed in 3 mL of PBS. After being cleaned, the samples were dried and photographed. The absorbance of the washed PBS samples was then determined using a UV spectrophotometer (Labman, Chennai, India) at 530 nm. The bacteria’s growth phase was measured using a UV-visible spectrophotometer at 660 nm to determine whether BN coating on the surface of Ti discs affected the growth rate of the microorganisms.

### 2.7. SEM Analysis

We used Zeiss FIB-FESEM (field emission scanning electron microscopy) with EDS (energy dispersion X-ray spectroscopy) for analysis. To prepare the samples for SEM, Karnovsky’s glutaraldehyde fixative (Himedia) was used for at least one hour, then 2% paraformaldehyde (Himedia), 2% glutaraldehyde (Himedia)**,** and 0.1 M phosphate buffer were applied for an additional hour. After that, post-fixation with osmium tetroxide and cacodylate phosphate buffer (Himedia) was done for an hour. Then samples were washed with deionized water. After that, a series of graded acetonitrile (Himedia) concentrations (50%, 70%, 90%, 95%, and 100%) was applied.

### 2.8. Statistical Analysis

First, data were entered into an Excel sheet. SPSS (Statistical Package for Social Sciences) version 21.0 was used to analyze the data. To determine the probability distribution of the data, the Kolmogorov-Smirnov test was used. It was discovered that the data were not normally distributed, therefore; non-parametric tests of significance were used. The Mann-Whitney U test was used to compare individuals from different groups. Median, interquartile range, and Z value were all measurements of interest. Statistical significance was defined as a *p* value < 0.05.

## 3. Results

### 3.1. Surface Characterization

Under the scanning electron microscope, a thin and uniform coating of 13–14 μm was observed at 25 KV accelerating voltage for the electrons. These surfaces were smooth, as evidenced by the profilometer’s roughness results. The uncoated and coated discs had different properties, discovered in the 0.11–3.44 μm range, which showed that the BN coating did not change the surface roughness. The hydrophobicity or hydrophilicity of implant surfaces was examined via contact angle measurements using distilled water. The contact angles of the implants ranged from 63 degrees to 79 degrees before BN coating. When coated with BN, however, all of the implants’ contact angles were between 46 and 67 degrees. This demonstrated how, despite its thinness, BN coating helped produce a hydrophilic surface with a similar contact angle. In addition, no cracks were found and the BN coatings showed good adhesions. Adhesion measurements were carried out to estimate a critical load; these adhesions could be seen clearly in SEM images.

### 3.2. Zone of Inhibition Test

Zone of inhibition tests done for *S. mutans* and *F. nucleatum* with control and BN-coated discs showed 11–13 mm in all BN-coated discs for *S. mutans*, *p* < 0.001 (Table 1). Although there was activity for *S. mutans*, there was no activity on control and *F. nucleatum* samples, which demonstrated *S. mutans* was susceptible to the action of BN-coated discs (Figure 2A and Figure 3).

### 3.3. Microbial Colony Forming Units (CFUs) Assay and Crystal Violet Staining Assay

Plates were counted after incubation for 48 h. In the microbial colony form in gun its (CFUs) assay, the colony forming units of *S. mutans* were significantly greater in group II samples [843 × 10^6^ (669.5 × 10^6^–964.25 × 10^6^)] as compared to those in group I [4.0 × 10^3^ (3.0 × 10^3^–7.25 × 10^3^)] (*p* value < 0.05) (Table 2, Figure 4A). The colony forming units of *F. nucleatum* were significantly greater in group II samples [555.5 × 10^6^ (534.0 × 10^6^–566.25 × 10^6^)] as compared to those in group I [540.0 × 10^3^ (451.0 × 10^3^–558.25 × 10^3^)] (*p*-value < 0.05) (Table 2, Figure 4B). Which demonstrates that BN showed bactericidal activity in this quantification method.

For the crystal violet assay, the washed PBS samples were examined in a UV spectrophotometer at 530 nm to determine their absorbance, while the growth phase of the bacteria was measured in a UV-visible spectrophotometer at 660 nm. The optical density of BN-coated discs was less in *S. mutans*, which means the growth phase of *S. mutans* bacteria was considerably less in BN-coated discs because more bacterial death occurred due to the physical characteristics of BN. However, the results against *F. nucleatum* were not significant (Table 3).

In the *S. mutans* samples, the optical density at 660 nm was significantly greater in group II samples as compared to that in group I samples [1.8540 (1.7290–1.9758) vs. 0.2725 (0.2558–0.2898)] (*p*-value < 0.05) (Table 3, Figure 5). Similarly, at 530 nm optical density was significantly greater in group II samples as compared to that in group I samples [0.6225 (0.5715–0.7090) vs. 0.0870 (0.0828–0.0890)] < 0.05) (Table 3, Figure 5).

In the *F. nucleatum* samples, the optical density at 660 nm was greater in group I [1.7230 (1.6280–1.7990)] as compared to that in group II [1.6970 (1.6250–1.7260)]; however, the difference was statistically non-significant (*p*-value > 0.05)) (Table 3, Figure 6). The optical density at 530 nm was greater in group II [1.0905 (1.0698–1.1280)] as compared to that in group I [1.0670 (1.0195–1.0908)], but this difference was also statistically non-significant (*p*-value > 0.05) (Table 3, Figure 6).

### 3.4. SEM Analysis Results

Since only *S. mutans* showed activity, we performed SEM for the same in uncoated and coated samples. Uncoated sample discs showed a mat layer of biofilm formed on the surface (Figure 7). In contrast, coated discs showed the characteristic nanospike-like structures of BN formed on the surface and showed maximum prevention of bacterial adhesion (Figure 8).

### 3.5. EDS Analysis Results:

At this point in the study, the analytical technique EDS (also called EDX or XEDS), was performed for elemental analysis and chemical characterization of the boron nitride coating on the titanium alloy discs. EDS results of BN-coated discs with and without microbial treatment showed marked results of BN coating %. (Figure 9 and Figure 10, Table 4) Carbon (C), oxygen (O), nitride (N), titanium (Ti), and boron (B) elements were all present. BN-coated discs with treatment showed the presence of 1.1% by weight B and 2.4% by weight N elements, and BN-coated discs without treatment showed the presence of 1.2% by weight B and 2.6% by weight N elements. 

## 4. Discussion

As the world’s population ages, the number of operations involving orthopedic implants for the hip and knee, as well as dental implants, is continuously rising. Dental implants in particular have been utilized for a wide range of purposes, including aesthetic treatment and the restoration of masticatory functions. Titanium and its alloys are frequently utilized as implant materials in orthopedic surgery and dentistry due to their superior mechanical qualities, great corrosion resistance, and biocompatibility [3]. However, there has also been an increase in TI implant failures caused by bacterial infections; such infections caused about 10% of Ti implant failures within a year of implantation. Additionally, it was observed that tissue inflammation and bone resorption were the main contributors to dental implant failures. Even though studies show that dental implants have a significant long-term success rate, biological complications still occur [33,34]. One of the most prevalent biological problems that can worsen and lead to implant failure is peri-implant disease [35]; therefore, reducing the incidence and severity of these diseases should be a top priority [36]. To stop initial bacterial adherence, several antibacterial coatings have been developed for TI. Despite some promising results using these coatings, newer implant methods are required due to the absence of long-lasting antibacterial effects and the emergence of drug-resistant microbes [37]. Previous studies have shown that TESPSA^27^, polydopamine, silver nanoparticles [28], BBF (a composite coating containing a new antibacterial agent (Z-)-4-bromo-5-(bromomethylene)-2(5H)-furanone) [38], as well as black phosphorus-zinc oxide nanohybrids [33], all have potential as antibacterial coatings for dental implant surfaces.

Recent studies have shown that BN and its derivatives, such as BN nanotubes, are less cytotoxic and more biocompatible [15,16,17]. Plasma treatment of BN nanotube films improved cell attachment; in addition, BN nanotube films increased the proliferation of human mammary cells [20,21], suggesting that these materials may have a significant potential for use in a variety of implant technologies. Coating over titanium implant surfaces can be done by physical vapor deposition, chemical vapor deposition, electrochemical deposition, 3-D printing, RF-magnetron sputtering, and spin coating methods [2]. A previous study demonstrated that the spin coating technique is a desirable method for thin film deposition: it is inexpensive, less risky, has remarkable controllability and reproducibility, is highly scalable, and offers considerable control over the properties of the generated films [23]. Therefore in the present study titanium alloy discs were coated with h-BNNSs using the spin coating method. 

Depending on the level of inflammation, peri-implantitis may result in the destruction of the alveolar bone. Peri-implantitis and periodontal diseases can be influenced by several bacteria species. According to a study, Gram-negative anaerobic species such as Prevotella, Streptococcus, Fusobacterium, and Treponema are the most prevalent bacteria found in peri-implantitis areas. Streptococcus mutans is the initial colonizer predominantly found on biofilm (more specifically, on a plaque). As a significant producer of biofilms, it is essential to prevent the formation of *S. mutans* on the surface of TI implants. Fusobacterium Nucleatum Gram-negative anaerobic is the most frequent bacteria found in peri-implantitis sites [31,32]. Thus in the present study, bactericidal activity against these two bacteria was evaluated in h-BNNSs coated discs compared with uncoated titanium discs. 

To check antimicrobial susceptibility, a zone of inhibition test was done for *S. mutans* and *F. nucleatum* with control and BN-coated discs. It showed activity for *S. mutans* but no activity for control and *F. nucleatum* samples, which means *S. mutans* is susceptible to the action of BN-coated discs (Table 1, Figure 1). In biofilm adhesion assays after incubation, colony counting, crystal violet staining, and UV spectrophotometer assay were performed. In the microbial colony forming units (CFUs) assay, the CFU of *S. mutans* and *F. nucleatum* were significantly greater in uncoated TI discs as compared to BN-coated discs. (Table 2, Figure 2) The colony forming units of *S. mutans* in BN-coated discs were significantly less, which demonstrated that in this quantification method, BN showed bactericidal activity.

In a crystal violet staining assay, adherent cells detach from cell culture plates during cell death. The amount of crystal violet staining in a culture decreases when cells that undergo cell death lose their adherence and are subsequently eliminated from the population of cells [24]. In the present study, after crystal violet staining samples were analyzed in a UV spectrophotometer at 530 nm to find the absorbance and at 660 nm to check the growth phase of the bacteria. The optical density of BN-coated discs was less in *S. mutans* (Table 3), which means the growth phase of *S. mutans* bacteria was considerably less in BN-coated discs because a greater number of bacterial deaths occurred due to the physical characteristics of BN (Figure 3). However, BN did not have a significant effect against *F. nucleatum* (Table 3, Figure 4).

In this study, Zeiss BUFIB FESEM with EDS analysis was used for the SEM analysis. Since *S. mutans* had shown prominent activity, SEM analysis was done to analyze the activity of *S. mutans* against BN-coated and uncoated discs. Uncoated discs showed a mat layer of biofilm on the disc surface (Figure 3). The BN-coated discs showed the nanospike-like structure of BN on the disc surface; they also showed maximum prevention of bacterial adhesion (Figure 4). EDS results showed a prominent release of boron and nitride elements: 1.1% by weight in B and 2.4% by weight in N elements were released from BN-coated discs with treatment. Without any treatment, BN-coated discs released 1.2% by weight of B and 2.6% by weight of N elements (Table 4, Figure 5 and Figure 6).

A previous study that quantified the biofilm adhesion and bacterial cell viability over titanium disc with or without antibacterial surface treatment using a silanization process like TESPSA demonstrated that peptides with a high surface density inhibited peptidoglycan biosynthesis and broke bacterial membranes [27]. Another study evaluated the antibacterial activity of titanium after surface modification with polydopamine and silver, and found that this coating effectively retarded microbial growth [28].

Previous in vitro studies have demonstrated that the physical interaction between the BN nanoflake-infused composite and the bacterial cellular envelope causes irreversible physical damage. Thus, BN-infused composite showed bactericidal activity [29,39,40]. Multiple antibacterial effects of graphene materials have been demonstrated in several investigations, including that after they penetrate the cellular envelope with sharp exposed edges, the bacterial cells undergo oxidative stress and wrap, which prevents nutrients from being transported across the membrane and results in death [3]. Specifically, it has been found that bacterial membrane integrity is lost and intracellular contents leak when bacteria come into direct contact with the sharp edges of graphene nanoparticles. Furthermore, the “chopping” effect caused by these sharp edges was found to significantly stress bacterial cell membranes [3].

Previous studies have shown that the sharpness and orientation of nanoparticles are critical elements to achieving significant bactericidal effects [3]. The current study found that because of BN’s spike-like structure, boron nitride nanosheet-coated titanium discs physically damaged the bacterial cell wall of *S. mutans* (an early colonizer) and helped to prevent initial biofilm formation. However, a limitation of the study was that BN was not able to significantly damage the cell wall of *F. nucleatum*. Uncoated discs also did not show any antibacterial action against *F. nucleatum*. However, in this study change in surface characteristics of titanium implants by BN coating did demonstrate significant antibacterial action. Thus, boron nitride nanosheet-coated titanium implants may be a promising antibacterial coating to help prevent both initial colonizers and biofilm formation, and thus reduce chances of peri-implantitis.

## 5. Conclusions

The present in vitro study aimed to compare the antimicrobial efficacy of boron nitride-coated titanium alloy discs with that of uncoated titanium alloy discs for dental implant applications. In this study, titanium alloy discs were coated with h-BNNSs through the spin coating method. Subsequently, zone of inhibition tests, microbial colony forming units assay, crystal violet assay, and SEM, along with EDS analyses, were done to check antibacterial efficacy.(1)Thin and uniform hydrophilic BN coating of 13–14 μm was achieved. In surface characterization, boron nitride showed good adhesion properties on the surface of titanium alloy discs.(2)Zone of inhibition tests done for *S. mutans* and *F. nucleatum* with control and BN-coated discs showed 11–13 mm in all BN-coated discs for *S. mutans*.(3)In microbial colony forming units (CFUs) assays, the CFUs of *S. mutans* and *F. nucleatum* were significantly greater in uncoated TI discs as compared to BN-coated discs.(4)In crystal violet assays, the growth phase of *S. mutans* bacteria was considerably less compared to *F. nucleatum* in coated discs. The optical density of BN-coated discs was less in *S. mutans*.(5)In SEM analysis, uncoated discs showed a mat layer of biofilm on the surface of the disc. Coated discs showed the characteristic nanospike-like structures of BN formed on the surface of the coating. This characteristic feature is responsible for bacterial cell wall damage.(5)EDS results showed a prominent release of boron and nitride elements. 1.1% by weight in B and 2.4% by weight in N elements were released from BN-coated discs with treatment. Without treatment, BN-coated discs released 1.2% by weight in B and 2.6 % by weight in N elements.


This in vitro investigation showed that boron nitride-coated discs were able to reduce bacterial adhesion and several microbial colonies in an in vitro biofilm compared to uncoated discs. However, it showed more prominent bactericidal activity against *S. mutans*; however, its antibacterial activity against *F. nucleatum* was not significant. Thus, boron nitride nanosheets may have potential as an antibacterial coating for titanium implants that will inhibit initial colonizer and biofilm growth, but further in vivo study needs to be done to assess future clinical implications.

## Figures and Tables

**Figure 1 jfb-14-00201-f001:**
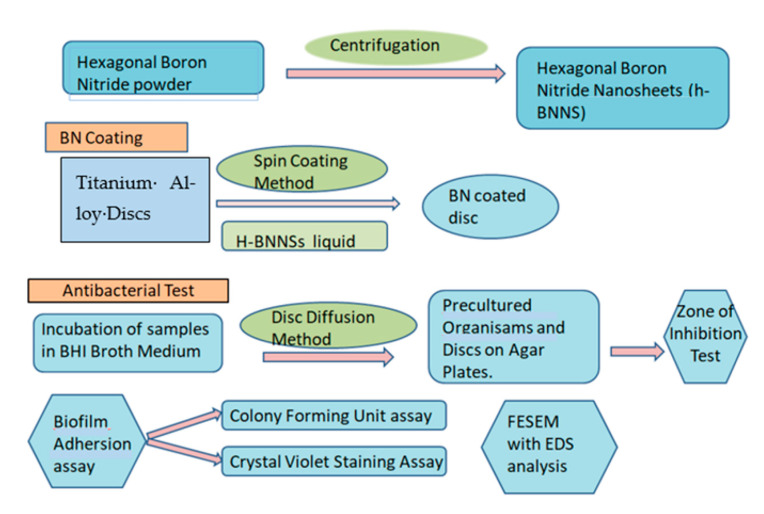
Schematic image of the experimental process.

**Figure 2 jfb-14-00201-f002:**
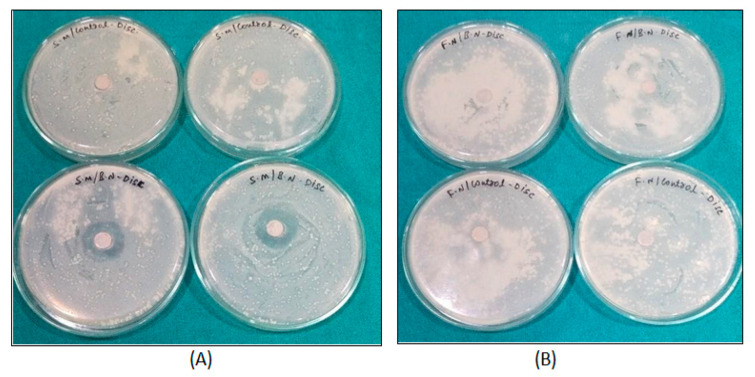
Zone of inhibition test for *S. mutans* (**A**) and *F. nucleatum* (**B**) with control and BN-coated discs.

**Figure 3 jfb-14-00201-f003:**
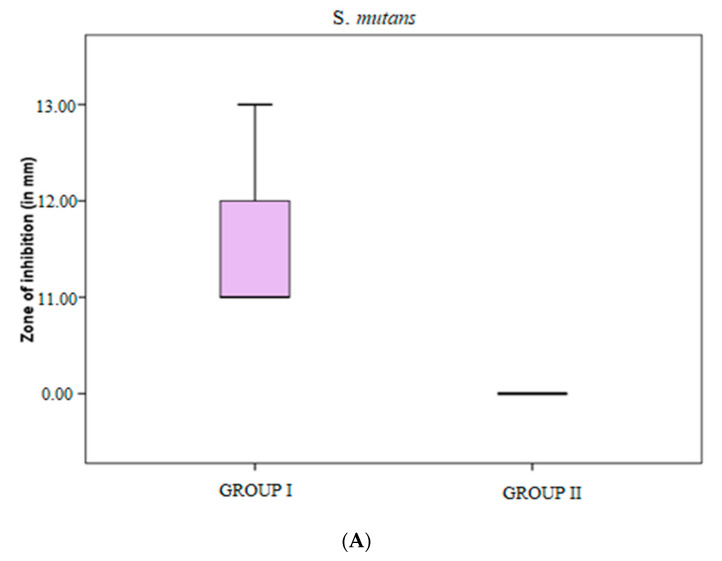
(**A**) Zone of inhibition against *S. mutans* in groups I and II. (**B**) Zone of inhibition against *F. nucleatum* in groups I and II.

**Figure 4 jfb-14-00201-f004:**
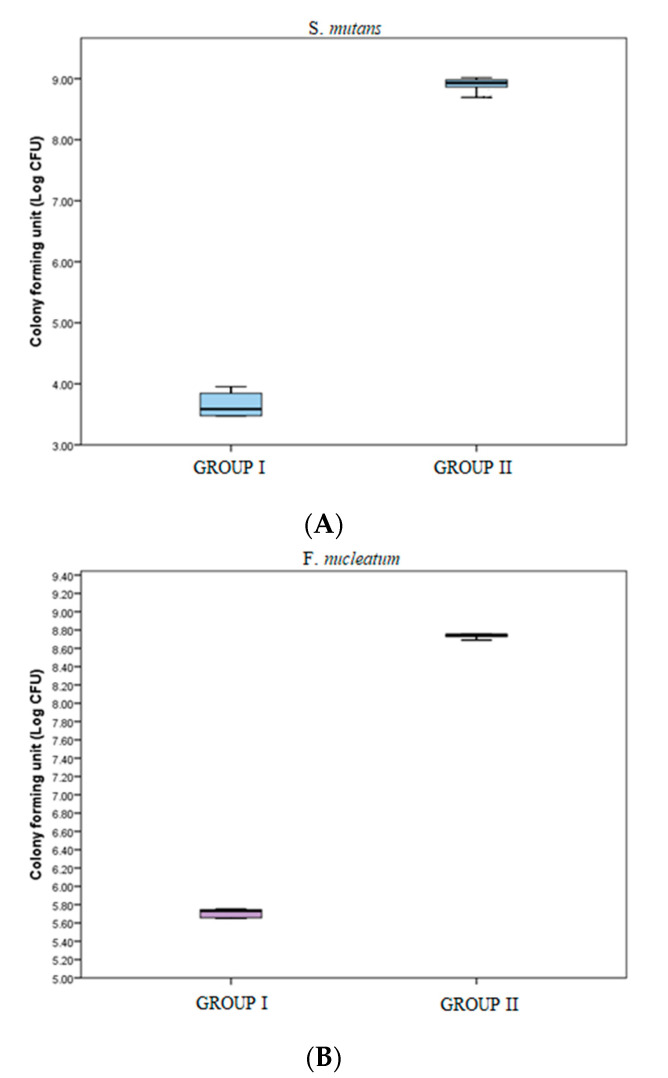
(**A**) Colony forming units (Log CFU) of *S. mutans* in groups I and II. (**B**) Colony forming units (Log CFU) of *F. nucleatum* in groups I and II.

**Figure 5 jfb-14-00201-f005:**
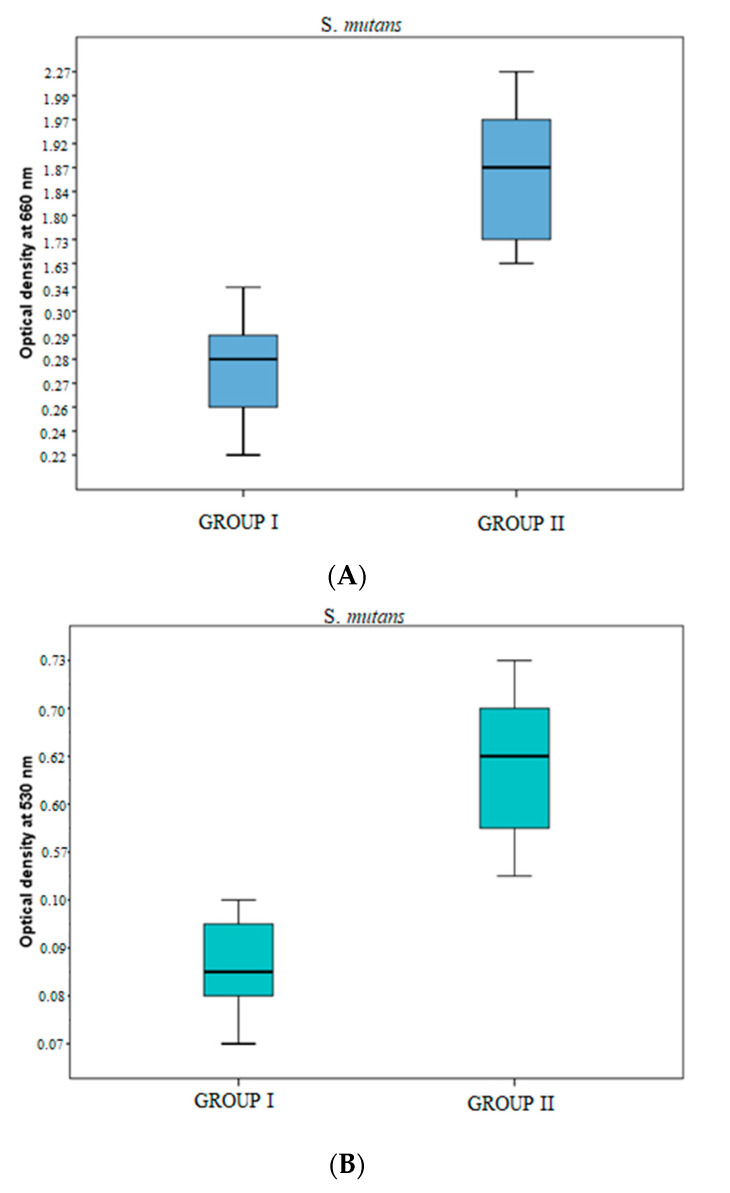
(**A**) Optical density of *S. mutans* samples at 660 nm in groups I & II. (**B**) The optical density of *S. mutans* samples at 530 nm in groups I & II.

**Figure 6 jfb-14-00201-f006:**
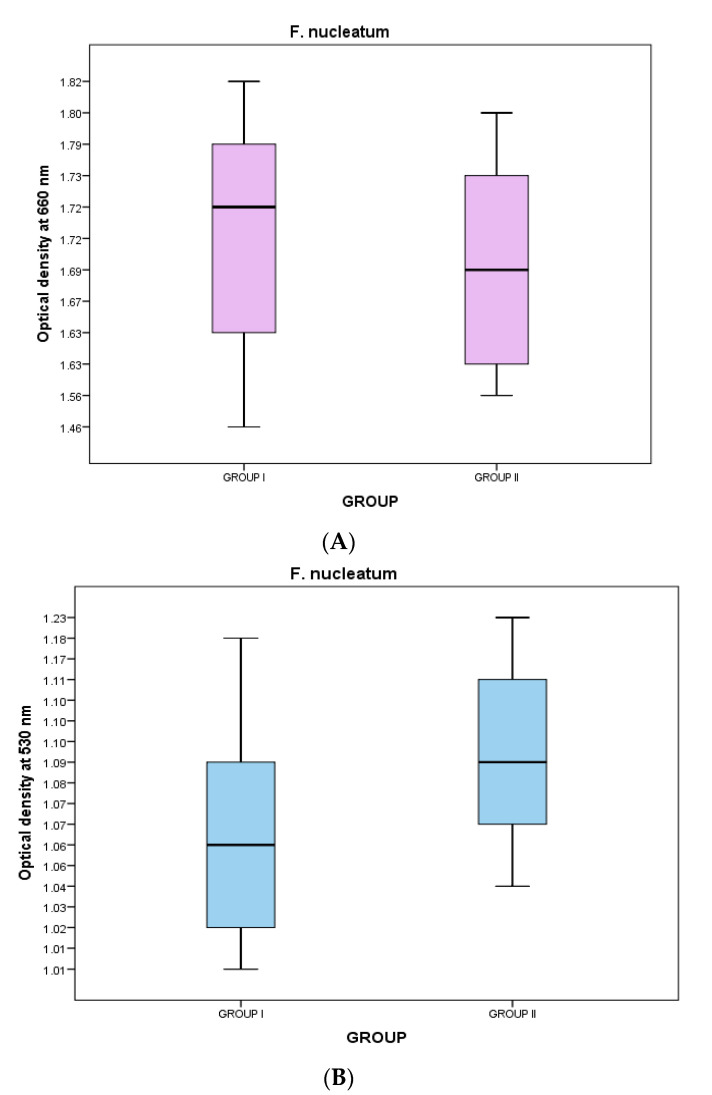
(**A**) Optical density of *F. nucleatum* samples at 660 nm in groups I and II. (**B**) The optical density of *F. nucleatum* samples at 530 nm in groups I and II.

**Figure 7 jfb-14-00201-f007:**
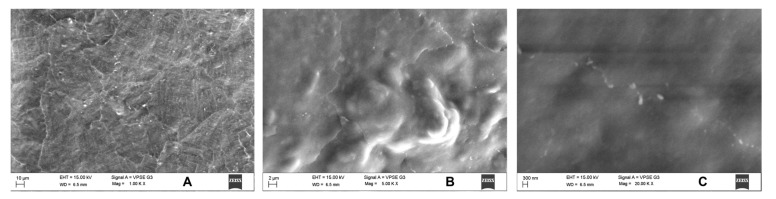
In SEM, uncoated discs showed a mat layer of biofilm formed on the disc surface. The samples were analyzed at (**A**) 1k×, (**B**) 5k× and (**C**) 20k× magnifications.

**Figure 8 jfb-14-00201-f008:**
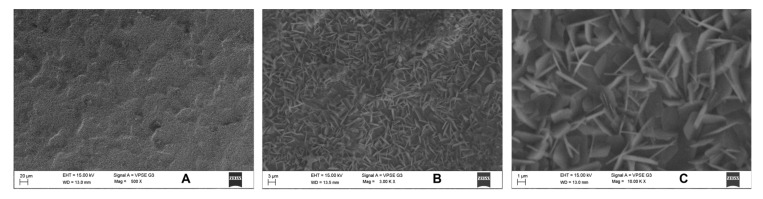
In SEM, BN-coated discs with characteristic structures of BN on the surface showed maximum prevention of bacterial adhesion. The samples were analyzed at (**A**) 500×, (**B**) 3k×, and (**C**) 10k× magnifications.

**Figure 9 jfb-14-00201-f009:**
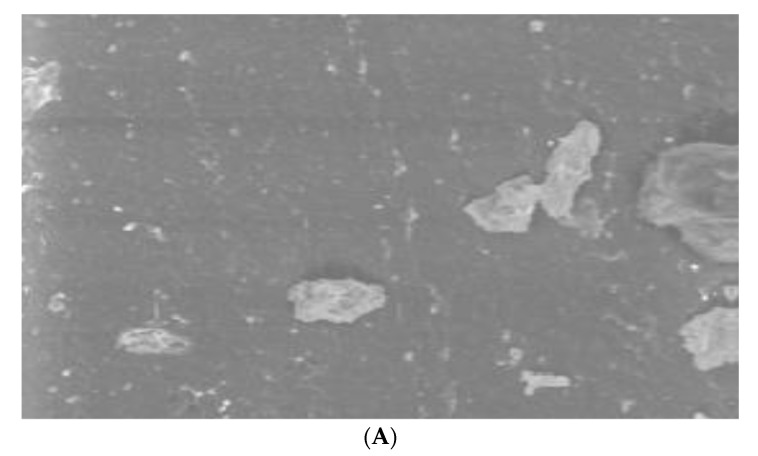
EDS analysis (**A**) Image at 500× magnification. (**B**) Figure of BN-coated discs with treatment.

**Figure 10 jfb-14-00201-f010:**
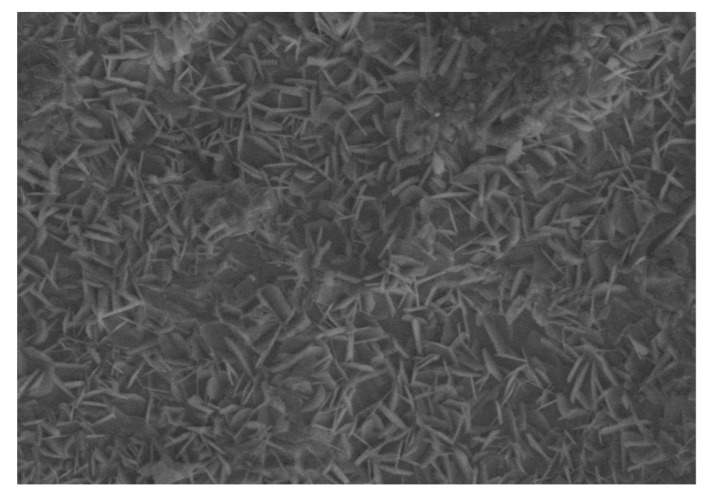
EDS analysis Image at 500× magnification.

**Table 1 jfb-14-00201-t001:** Comparison of zone of inhibition in groups I and II for *S. mutans* and *F. nucleatum*.

Microorganism	Disc Type		Zone of Inhibition (mm)		Z Value	*p* Value ª
			Median	Interquartile Range		
*S. mutans*	Group I		11.0	11.0–12.0	4.108	0.001 *
	Group II		0.0	0.0–0.0		
*F. nucleatum*		Group I	0.0	0.0–0.0	-	-
		Group II	0.0	0.0–0.0		

ª Mann-Whitney U test. * *p* value < 0.05 was considered statistically significant.

**Table 2 jfb-14-00201-t002:** Comparison of colony forming units in groups I and II for *S. mutans* and *F. nucleatum*. ª Mann-Whitney U test. * *p* value < 0.05 was considered statistically significant.

	Group			Colony Forming Unit		Z Value	*p* Value ª
				Median	Inter-Quartile Range		
*S. mutans*	Group I			4.0 × 10^3^	3.0 × 10^3^–7.25 × 10^3^	3.808	0.001 *
	Group II			843 × 10^6^	669.5 × 10^6^–964.25 × 10^6^		
*F. nucleatum*		Group I	540.0 × 10^3^		451.0 × 10^3^–558.25 × 10^3^	−3.784	0.001 *
		Group II	555.5 × 10^6^		534.0 × 10^6^–566.25 × 10^6^		

**Table 3 jfb-14-00201-t003:** Comparison of optical density in groups I and II for *S. mutans* and *F. nucleatum*.

*S. mutans*	Group	Optical Density		Z Value	*p* Value ª
		Median	Interquartile Range		
Optical density at 660 nm at 660 nm	Group I	0.2725	0.2558–0.2898	−3.784	0.001 *
	Group II	1.8540	1.7290–1.9758		
Optical density at 530 nm at 530 nm	Group I	0.0870	0.0828–0.0890	−3.787 −3.787	0.001 *
	Group II	0.6225	0.5715–0.7090		
** *F. nucleatum* **	**Group**	**Optical Density**		**Z Value**	***p* Value ª**
		**Median**	**Interquartile Range**		
Optical density at 660 nm	Group I	1.7230	1.6280–1.7990	−0.835	0.436
	Group II	1.6970	1.6250–1.7260		
Optical density at 530 nm	Group I	1.0670	1.0195–1.0908	−1.817	0.075
	Group II	1.0905	1.0698–1.1280		

ª Mann-Whitney U test. * *p* value < 0.05 was considered statistically significant.

**Table 4 jfb-14-00201-t004:** EDS analysis results for coated elements with and without treatment.

Elements with Treatment	Weight %	Atomic %	Error %
C K	49.8	78.2	11.7
O K	15.9	21.2	12.9
N K	2.4	0.35	29.1
Ti K	30.8	1.6	7.5
B K	1.1	0.9	23.3
**Without Treatment**	**Weight %**	**Atomic %**	**Error %**
C K	50.4	77.2	10.2
O K	15.6	21.6	12.8
N K	2.6	0.4	27.8
Ti K	30.2	1.2	7.5
B K	1.2	0.9	24.3

## Data Availability

The data set used in the current study will be made available at any reasonable request.

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
