# Peer review of "Antibacterial Efficacy and Surface Characteristics of Boron Nitride Coated Dental Implant: An In-Vitro Study"

_jfb, 2023, doi:10.3390/jfb14040201_

Round 1

Reviewer 1 Report

Sample size analysis missing. 
More in depth explanation needed.
Graphs and figures should be more readable.
Discussion should be expanded.

Author Response

To,

The Reviewer ,JFB.

Subject: Regarding  correction in  the manuscript  entitled” Antibacterial efficacy and surface characteristics of boron nitride coated dental implant. An in-vitro study”

Respected sir

We would like to explicitly state that we agree with all the comments as these helped us improve the quality of our paper. We have made a conscious effort to answer all the remarks in the paper as advised by the reviewers and highlighted changes with track changes in the revised manuscript for their convenience.

Sample size analysis missing. 
More in depth explanation needed.
Graphs and figures should be more readable.
Discussion should be expanded.

Response to Reviewer’s comments:

20 Sample size was taken by considering the budget of the research work  and after doing analysis.

Necessary correction in discussion has been done.

Increased and clear size of Graphs and figures has been added to make it more readable.

Thanking You.

Reviewer 2 Report

1. In line 47, literature should be referred to prove“A dental implant's success or failure can be greatly influenced by osseointegration, which occurs when an implant integrates with the bone, and bacterial aggregation sur-  rounding the implant” as well as implant-related infections in particular might result in major consequences”.

2. In line 53 literature should be referred to prove “Integration of the dental implant material into the surrounding bone and connective tissue is important for the long-term stability of dental implant material”.

3.  At the end of Introduction, you need to write down the important aim(s) that you are doing this research for because it is not stated at all.

4. Importantly, although the surface modification of implants has beneficial effects on antibacterial properties, the author has not mentioned whether such modification have an impact on osseointegration. Author must clarify it.

5. Limitation needs to be clarified in Discussion.

6. Figure 1 needs to be more concise and clear

7. The capitalization in Figure 1 is inconsistent and needs to be modified.

8. Figure 2 The picture quality is too poor. Each group of pictures needs to be taken separately and clearly marked with the invasion zone.

Author Response

To,

The Reviewer ,JFB.

Subject: Regarding  correction in  the manuscript  entitled” Antibacterial efficacy and surface characteristics of boron nitride coated dental implant. An in-vitro study”

Respected madam/sir,

We would like to explicitly state that we agree with all the comments as these helped us improve the quality of our paper. We have made a conscious effort to answer all the remarks in the paper as advised by the reviewers and highlighted changes with track changes in the revised manuscript for their convenience.

  1. In line 47, literature should be referred to prove“A dental implant's success or failure can be greatly influenced by osseointegration, which occurs when an implant integrates with the bone, and bacterial aggregation sur-  rounding the implant” as well as “implant-related infections in particular might result in major consequences”.
  2. In line 53 literature should be referred to prove “Integration of the dental implant material into the surrounding bone and connective tissue is important for the long-term stability of dental implant material”.

Response: Thank you for your insightful suggestion, reference in line 47 and 53 have been added in the revised manuscript.

  1. At the end of Introduction, you need to write down the important aim(s) that you are doing this research for because it is not stated at all.

Response: Thank you for your insightful comments and suggestion, the aim of the study has been explained in separate paragraph at the end of introduction in the revised manuscript.

  1. Importantly, although the surface modification of implants has beneficial effects on antibacterial properties, the author has not mentioned whether such modification have an impact on osseointegration. Author must clarify it.

Response: Thank you for your insightful comments and suggestion, correction have been carried out in the revised manuscript.

  1. Limitation needs to be clarified in Discussion.

Response: Limitation of study has been explained in the discussion section as per your insightful comments and suggestion

  1. Figure 1 needs to be more concise and clear
  2. The capitalization in Figure 1 is inconsistent and needs to be modified.
  3. Figure 2 The picture quality is too poor. Each group of pictures needs to be taken separately and clearly marked with the invasion zone.

Response: Figure size and quality has been increased, as per your insightful comments and suggestion

Author Response

To,

The Reviewer ,JFB.

Subject: Regarding  correction in  the manuscript  entitled” Antibacterial efficacy and surface characteristics of boron nitride coated dental implant. An in-vitro study”

Respected madam/sir,

We would like to explicitly state that we agree with all the comments as these helped us improve the quality of our paper. We have made a conscious effort to answer all the remarks in the paper as advised by the reviewers and highlighted changes with track changes in the revised manuscript for their convenience.

Response to Reviewer’s comments

 Correction in the manuscript spelling and grammar has been done.

The aim of research has been explained in the last paragraph in the introduction.

Regarding use of titanium alloy correction has been done in the introduction and figure 1

Since this the first research conducted on coating of boron nitride over titanium alloy disc with spin coating method. In discussion Spin coating method and its importance has been given with reference. As it has already been proven as most frequently used coating method. Regarding coating uniform coating has been achieved surface characterization tests has been done and explained in result In SEM analysis 11-13 Micrometer coating with characteristic spike like structure is visible in the given Picture. XRD analysis could not be done due to lake of this facility and financial constrain.

EDS analysis has been done to evaluate number of elements released of BN coated surface  after doing this analysis chemical characteristics of BN could achieved with appropriate release of Boron and Nitride Elements in percentage.

Detail process of evaluating antibacterial effect has been explained in point 2.2 to 2.6 other material which has been used as antibacterial agents like TESPRA and silver nanoparticles all explained in the discussion section with reference

Conclusion has been modified as per requirement.

Kindly go through submitted manuscript.

Thanking You.

Round 2

Reviewer 2 Report

I would like to thank and appreciate the authors for their efforts to correct the manuscript.

Author Response

Dear Prof,

Thank you so much for your kind words, highly appreciated

Best regards

Reviewer 3 Report

The Dr. Mohmed Karobari manuscript is devoted to evaluated the antibacterial effect and surface to pography of boron nitride-coated implants to prevent pellicle formation and bacterial colony formation, hence reducing the chances of periimplantitis. This work combines complex lab research from two science remote areas – nanomaterials, biomedicine and dental. It should be noted that a lot of lab work has been done. However, in this form the manuscript cannot be published and needs to be improved:

1.        Authors must respond to each of my comments. Point by point.

2.        The formatting of the manuscript is still unacceptable. The authors did not adhere to the requirements of the MDPI, the figs are of poor quality, extra spaces, dots, commas, they are missing somewhere, and so on. For example fig5 and fig6 cannot be seen.

3.        Poor reference list.

4.        English is still poor. Please work on it. To be more precise, the language is better, but the style leaves much to be desired. For example, “Here in this study.....”, repetition of conjunction "that". The use of personal pronouns is not recommended.

5.        Add a couple of links to similar materials that are used in this area, such as porous alumina: Dental composites with strength after aging improved by using anodic nanoporous fillers: experimental results, modeling, and simulations / A. Ghorbanhossaini, R. Rafiee, A. Pligovka, M. Salerno // Springer, Engineering with Computers 2022. https://doi.org/10.1007/s00366-021-01566-6

6.        Line 32:Pure titanium still appears in the abstract.

7.        To apply h-BN, a technique consisting of several operations was used. However, the authors did not show what was eventually formed on the surface of the alloy. To do this, they could use for example XRD or give a link to studies where a film of boron nitride was formed using the same technique.

8.        Figs 5 and 6 show EDS after bacterial treatment. It is not clear why this was done and what results the authors expected to get?

Now it is clear that you expected a decrease in BN under the influence of processing.

Line 436-439 The decrease is only 0.1% for B, and 0.2% for N. What is the accuracy of your EDS? What I see in the error column of Table 4 does not allow you to draw such conclusions.

9.        Please describe in more detail and clearly the process of evaluating the antibacterial effect. Provide a comparison with other materials if you claim high efficiency of BN.

10.     The conclusion should consist of two parts, in the first, briefly describe what was done, in the second, point by point, list the main results obtained, on which the main conclusion is made:

So-and-so is done, so-and-so is got. As a result of the work, the following points have emerged:

1.

2.

3.

4.

Author Response

Authors would like to thank the reviewer for taking precious time to review this manuscript and give us comments. We would like to explicitly state that we agree with all the comments as these helped us improve the quality of our paper. We have made a conscious effort to answer all the remarks in the paper as advised by the reviewer and highlighted changes with track changes in the revised manuscript for their convenience.
